# Effect of glycemic control and type of diabetes treatment on TB treatment outcomes among people with TB-diabetes: A systematic review (updated August 2024)

Hemant Deepak Shewade[1,2☯*], Prabhadevi Ravichandran[1☯*], S. Satish[1], S. Kiran Pradeep[1,3], Kathiresan Jeyashree[1,2], Preetam Mahajan[4], Amar N. Shah[5], Richard Kirubakaran[6], Ajay M. V. Kumar[7,8]

1 ICMR – National Institute of Epidemiology (ICMR-NIE), Chennai, India, 2 Academy of Scientific and Innovative Research (AcSIR), Ghaziabad, India, 3 Madha Medical College and Research Institute, Chennai, India, 4 Jawaharlal Institute of Postgraduate Medical Education & Research (JIPMER Karaikal), Puducherry, India, 5 United States of America Agency for International Development (USAID), American Embassy, New Delhi, India, 6 Centre for Biostatistics and Evidence-Based Medicine (CBEBM), Vellore, India, 7 The Union South-East Asia Office, New Delhi, India, 8 Yenepoya Medical College, Yenepoya (Deemed to be University), Mangaluru, India.

☯ Contributed equally as primary authors.
* deviprabha28@gmail.com (PR), hemantjipmer@gmail.com (HDS)

## Abstract

### Background

Stringent glycemic control and/or using insulin either as a replacement for or in addition to oral hypoglycemic agents (OHAs) has been recommended for people with tuberculosis and diabetes mellitus (TB-DM). This systematic review (PROSPERO 2016:CRD42016039101) analyses whether this improves TB treatment outcomes. This is an updated review (up to August 2024) of a previously published systematic review (1996 – April 2017).

### Objectives

Among people with drug-susceptible TB-DM on anti-TB treatment, to determine the effect of i) glycemic control (stringent or less stringent) compared to poor glycemic control and ii) insulin (only or with OHAs) compared to 'OHAs only' on unfavorable TB treatment outcome(s) at the end of intensive phase and/ or end of TB treatment (minimum six months and maximum 12 months follow up).

### Methods

We conducted comprehensive searches across multiple databases (EMBASE, PubMed, Google Scholar, Cochrane Database of Systematic Reviews) and sources. Eligible studies included interventional and cohort studies examining people with

**Data availability statement:** All relevant data are within the paper and its Supporting Information files.

**Funding:** The author(s) received no specific funding for this work.

**Competing interests:** The authors have read the journal's policy and have the following competing interests: Hemant Deepak Shewade is a member of the PLOS Global Public Health Editorial Board.

TB-DM. Screening, data extraction and risk of bias assessment were done independently by two investigators and recourse to a third investigator, for resolution of differences.

## Results

From a total of 7107 articles, we included 14 studies, with five added in this update (all observational cohort studies). Of 14, only one high-quality study reported that stringent glycemic control (HbA1c<7% at baseline) was associated with lower risk of unfavorable treatment outcomes, including recurrence, compared to non-stringent and/or poor glycemic control. Other studies showed mixed results and had significant biases or were limited by sample size. The five newly included studies had a high risk of bias and did not provide clear evidence. Due to clinical and methodological heterogeneity, we did not perform a meta-analysis.

## Conclusion

The updated review re-emphasizes the need for high-quality research on the effects of glycemic control and addition of insulin among people with TB-DM on TB treatment outcomes. We need well-designed randomized controlled trials, specifically for the effect of adding insulin on TB treatment outcomes. We discuss ten measures to guide well-designed cohort studies on this topic. Harmonization of the methods is needed and would facilitate comparisons.

## Introduction

Tuberculosis (TB) and diabetes mellitus (DM) represent two critical challenges to global health, where each or both combined significantly affects the individuals, communities and healthcare systems worldwide [1,2]. TB continues to be a major contributor to illness and death, with an estimated 10 million incident cases and 1.4 million TB (related) deaths reported in 2022 [1]. The burden of DM is consistently increasing, especially in low- and middle-income countries [2].

An estimated 0.37 million annual incident TB is attributed to DM [3]. There exists a bidirectional association between TB and DM as they complicate one another's pathogenesis and treatment [4–7]. DM further compromises the already weakened immune system of people with TB, leading to poor blood sugar control, a higher likelihood of treatment failure, and an increased risk of death during TB treatment (when compared to those without DM). People with TB-DM are two to three times more likely to remain culture-positive, four times more likely to experience a relapse after completing treatment, and five times more likely to die compared to those without DM [8–13]. Conversely, TB treatment complicates the management of glucose levels, thus diminishing the effectiveness of DM management [14]. This association has become a major concern among clinicians and public health physicians.

Among people with TB-DM, international guidelines recommend keeping HbA1c levels below 8% or capillary fasting blood glucose (cFBG) under 10 mmol/l (180 mg/dl) during TB treatment. This is in line with DM management in persons with a significant co-morbidity [15]. Glycemic control targets among non-pregnant adults (without comorbidities) are recommended as follows: HbA1c<7% or cFBG/ pre-prandial capillary blood glucose <7.2 mmol/l corresponding to 130 mg/dl or peak post prandial capillary blood glucose <10 mmol/l corresponding to 180 mg/dl [16]. The benefit and the potential risks of achieving these more stringent glucose control targets among people with TB-DM are currently unknown [15].

Metformin is the preferred oral glucose-lowering medication for people with TB [15]. Insulin is recommended for severe hyperglycemia (HbA1c over 10% or FBG over 15 mmol/l or 270 mg/dl) and uncontrolled blood glucose levels with oral hypoglycemic agents (OHAs) [15,17]. There is ongoing debate regarding whether insulin offers significant advantages over OHAs in improving TB treatment outcomes. Clinicians suggest potential benefits in using insulin for glycemic control and improved TB outcomes [18]. Non-availability of insulin in resource-limited settings, lack of compliance, emergence of adverse effects, risk of hypoglycemia and the need for self-monitoring of blood sugar levels, restrict the usage of insulin in real-world settings [19].

Previously, in a published systematic review involving studies until 25 April 2017, we identified few studies that provided quality information on effect of glycemic control and/or insulin (with or without oral hypoglycemic agents compared to oral hypoglycemic agents alone) on TB treatment outcomes among individuals with TB and DM (TB-DM) [20]. The review also highlighted the need for robustly designed and analyzed studies including randomized controlled trials on the effect of glucose lowering treatment options on TB treatment outcomes [20]. Since the publication of review, the evidence remains inconclusive, with some studies reporting conflicting results [21,22]. There has been a gradual increase in the number of studies examining the impact of glycemic control and glucose lowering treatment on TB treatment outcomes. We expect new studies to be of methodological rigor, improved adjustment of confounding variables and probably a randomized controlled trial, which was lacking in previous review. Therefore, this systematic review aims to update the existing review thereby offering greater clarity, addressing previous knowledge gaps and generate updated evidence on this topic [20].

The objectives remained the same as in the previous systematic review, to synthesize the evidence (till August 2024) on the effect of different glycemic control strategies (stringent, less stringent, and poor glycemic control) and DM treatment approaches (insulin only, insulin with OHAs, and OHA only) on TB treatment outcomes in individuals with TB-DM **(primary objective)** [20]. Additionally, we evaluated the development of multi-drug-resistant TB (MDR-TB, if drug-susceptible at baseline), recurrence rates after completing treatment, and the occurrence of adverse events related to glucose-lowering therapies **(secondary objective)** [20].

## Methods

### Protocol

The protocol was registered with PROSPERO (PROSPERO 2016: CRD42016039101) and can be accessed online [23]. This update maintains adherence to the original protocol with no deviations.

### Inclusion criteria

**Study type.** In the previous review, all interventional (with a control arm) and cohort studies from 1996 to 25 April 2017 were included, regardless of country or language [20]. We extended this to include studies from 26 April 2017 to 31 August 2024. Excluded were single-arm interventions, before-after designs without control, cross-sectional studies, case-control studies, case series, case reviews, subject reviews and ecological studies.

**Participant characteristics.** Individuals of any age and gender who have TB and are on anti-TB treatment (ATT – not known to be drug-resistant at baseline) and who have been diagnosed with DM (type I, II, or other) either before or during the TB diagnosis or during ATT. Characteristics included various ATT regimens, HIV status, healthcare settings

(programmatic/ clinical/ public/ private/ inpatient/ outpatient), TB types (pulmonary/ extrapulmonary), diagnostic methods (microbiologically/ clinically diagnosed TB), and previous treatment (new/ retreatment TB).

**Interventions/exposure.** For the first primary objective, 'stringent glycemic control' was characterized by cFBG < 130 mg/dl, capillary post-prandial blood glucose <180mg/dl or HbA1c < 7%. 'Less stringent targets' were defined as cFBG < 180mg/dl, capillary post-prandial blood glucose <206 mg/dl or HbA1c < 8% [16]. Glycemic control could be evaluated at different stages of TB treatment (baseline/ before or after the intensive phase). The comparison group consisted of individuals with poor glycemic control; cFBG ≥ 180 mg/dl, capillary post-prandial blood glucose ≥206 mg/dl or HbA1c ≥ 8%. For the second primary objective, participants on 'insulin only' or 'insulin with OHAs', were compared with people receiving 'OHAs only'.

**Outcome.** In line with World Health Organization (WHO) recommendation, primary outcomes comprised favorable (sputum conversion at the end of intensive phase, cure or treatment completed at the end of treatment) or unfavorable (all other outcomes) TB treatment outcomes at the end of intensive phase and/or TB treatment [24]. Secondary outcomes included the development of MDR-TB during treatment, recurrence after treatment, and adverse events related to glucose-lowering therapy.

## Search strategy

Electronic searches were conducted across several databases including PubMed, EMBASE, Google Scholar, and the Cochrane Database of Systematic Reviews (S1 and S1A appendix). Additionally, efforts were made to identify relevant studies through cross-referencing within included studies, consulting experts (ongoing or recently completed on this topic) and thoroughly searching grey literature sources. These sources included ISI Web of Science, ClinicalTrials.gov, the WHO International Clinical Trials Registry Platform (ICTRP) search portal (apps.who.int/trial search/), meta Register of Controlled Trials (controlledtrials.com/mrct/) and trial results registers, and guidelines and their reference lists.

## Data collection and analysis

Screening, data extraction, and assessment of bias all were independently performed by two investigators (PR and SKP/ HDS). Any disagreements were resolved through consensus between the investigators, with the involvement of a third investigator (SS), if necessary.

**Study selection.** Duplicates were removed using Mendeley software, and the bibliography was imported into Rayyan for initial screening based on titles and abstracts. During full-text screening, studies were included for data extraction only if they met all criteria (study type, participant characteristics, intervention/exposure with comparator, and outcomes).

**Data extraction and management.** We extracted study data from the full text of the included studies into a data extraction form (S2 and S2A appendix). If primary data were unavailable in the supplementary files and/or no effect measures (adjusted/unadjusted) were reported in line with our review objectives, we requested the same from the authors. We sent three email reminders, fortnightly and awaited response for a maximum period of 1 month after writing the first email to them.

**Assessment of bias in included studies.** For interventional studies, we planned to assess the risk of bias in the following domains: sequence generation, allocation concealment, blinding of participants, personnel, and outcome assessors, incomplete outcome data, selective outcome reporting, and other sources of bias such as carry-over, recruitment bias, and contamination. We planned to rate the studies as high, low, or unclear risk of bias in each domain [25]. For cohort studies, we utilized a modified New Castle-Ottawa quality assessment scale and presented the results in a risk of bias table [26].

**Measures of effect.** We intended to report the effect separately for interventional and cohort studies using adjusted odds ratio, relative risk, hazard ratio (aOR, aRR, aHR) or crude (cOR, cRR, cHR) measures along with 95% confidence intervals (CI) and conduct meta-analysis in the absence of clinical, methodological and statistical heterogeneity. If



adjusted comparisons were not provided by the authors, we planned to calculate crude effects based on available data from narrative texts or tables. In cases where primary data were accessible, we aimed to conduct analyses after appropriate adjustment and potentially perform a pooled analysis, if feasible.

**Assessment of reporting biases.** If enough studies were included (at least 10) in meta-analysis, we intended to use funnel plots to detect publication bias. We adhered to the Preferred Reporting Items for Systematic Reviews and Meta-Analyses (PRISMA) statement and the Meta-analysis Of Observational Studies in Epidemiology (MOOSE) checklist for reporting this review [27,28].

## Results

### Search findings

Between 1996 and August 2024, we identified 7107 published articles. Following the removal of duplicates, 5673 articles underwent title and abstract screening, and 107 were assessed for eligibility during full-text screening (Fig 1). Details on the reasons for excluding 93 studies during full-text screening are provided in S1 Annex. Fourteen studies were included for qualitative assessment and data extraction (Fig 1) and of these five were between 2017 and 2024 (Kornfeld H et al (2020), Udaykumar P et al (2022), Baltas I et al (2023), Kim KH et al (2024) and J Mistry et al (2024) [22,29–32].

### Description of included studies

Of the 14 studies included in the review (see S2 Annex for characteristic of included studies), all were cohort studies: 13 focused on glycemic control and its effect on TB treatment outcomes (Table 1) [22,29–40] and six examined the effects of different DM treatments on TB treatment (Table 2) [22,29,30,32,35,41]. All studies included both sexes and adults, one study had predominantly male and one study included children [31,36]. It remains unclear whether children were included in two other studies [33,39]. There was no mention of patient characteristics (age or sex) in one study [32]. Five studies involved people with and without HIV [30,33,35,37,40], while five studies specifically excluded people living with HIV [22,29,34,38,39]. In the remaining four studies, information on HIV status was not provided [31,32,36,41].

cRRs were derived from narrative texts and tables. As there was clinical and methodological heterogeneity, meta-analysis was not done and a funnel plot to detect publication bias was also not generated [39]. All the five studies included in this update (between 2017 and 2024) provided cRR upon request and shared previously unreported outcomes or shared outcomes as per the review's requirement (one provided aRR and one aOR) [22,29–32].

### Risk of bias assessment

Of the 14 included studies, only one study by Mahishale et al did not have any risk of bias (see Table 3 and S2 Annex for details) [39]. All other studies had more than one risk of bias or at least one major bias. Due to substantial risk of bias and clinical and methodological heterogeneity, we presented the results of these 14 studies qualitatively (Fig 1). Clinical heterogeneity includes differences in populations (pulmonary or extrapulmonary TB, presence or absence of HIV). Methodological heterogeneity includes major differences in interventions/ exposure (varying cut-offs and tests to assess glycemic control at different points in TB treatment) across studies. Additionally, some studies used poor glycemic control as reference (HbA1C more than 8%) while some studies used non-stringent and/or poor control as reference (HbA1C more than 7%).

### Glycemic control and TB treatment outcomes

There were a total of thirteen (eight till April 2017 and five from this updated review) studies that investigated the effect of glycemic control on TB treatment outcomes (Table 1) [22,29–40]. Out of them, three studies reported that glycemic control was associated with lower risk of unfavorable outcomes when compared to those without glycemic control. An Indian study



Records identified through EMBASE, PubMed, Google Scholar and other database search

(N= 7107)

↓

Records after removal of duplicates and subjected to title/abstract screen

(N=5673)

→ Studies excluded after title/abstract screen (N=5555)

Included after title/abstract screen but excluded due to non-availability of full text despite contacting the authors (N=11)

↓

Studies assessed for eligibility (full text screen) (N=107)

→ Studies excluded (N=93)

Reasons for exclusion:

- Outcome not fulfilled- 15
- Exposure/comparator not fulfilled- 62
- Participant criteria not fulfilled- 9
- Type of study not fulfilled- 5
- Type of study not fulfilled and exposure comparator not fulfilled- 2
- Participant criteria and exposure comparator not fulfilled- 10
- Exposure comparator and outcome not fulfilled- 7
- Participant criteria, exposure comparator and outcome not fulfilled- 5

↓

Studies included for qualitative assessment

(N=14)

→ High clinical and methodological heterogeneity

- Different settings
- Different patient subtypes
- Different diagnostic/follow up tests and cut offs
- Different ways of assessing glycemic control during TB treatment
- Different ways of analysis including inappropriate measures of association and non-availability of adjusted associations for the desired comparisons.

↓

Studies included for meta-analysis
(N=0 )

**Fig 1. PRISMA flow diagram for the systematic review conducted between 1996 to August 2024.**



**Table 1. Effect of glycemic control (stringent or less stringent) on unfavorable TB treatment outcomes, summarized as crude/adjusted relative risk/ odds ratio (cRR/ aRR/ cOR/ aOR) @*.**

| Study ID | Reference group | Exposed group | cRR* | (95% CI) |
|---|---|---|---|---|
| **All unfavorable end (TB) treatment outcome** | | | | |
| Chiang CY_2015_Plos One | Poor glycemic control (HbA1c > 9) | Less stringent glycemic control (HbA1c 7–9) | 1.53 | (0.96, 2.46) |
| | | Stringent glycemic control (HbA1c < 7) | 1.89 | (1.12, 3.20) |
| Mi F_2013_TMIHˆ | Poor glycemic control (FPG > 10mmol/l) | Less stringent glycemic control (FBG 7–10 mmol/l) | 0.91 | (0.18,4.43) |
| | | Stringent glycemic control (FPG < 7mmol/l) | 1.03 | (0.21,5.07) |
| Magee MJ_2013_International J of Infectious diseases | Poor glycemic control (no specific criteria) | Glycemic control | 1.08 | (0.54, 2.16) |
| Nandakumar KV_2013_Plos One | Poor glycemic control (FBG > 100mg/dl and PPBS/RBG > 140mg/dl) | Glycemic control | 0.52 | (0.25, 1.07) |
| Tabarsi P_2014_Journal of Diabetes and Metabolic disorder | Poor glycemic control (HbA1c≥6.5) | Glycemic control | 1.13 | (0.20,6.44) |
| Yoon YS_2017_ Thorax | Poor glycemic control HbA1c≥7) | Glycemic control | 0.55 | (0.22, 1.36) |
| Mahishale V_2017_Iran J MS | Poor glycemic control HbA1c≥7) | Glycemic control | 0.18 | (0.09, 0.36) |
| *Kornfeld H_2020_Clinical infectious diseases* | *Poor glycemic control (HbA1c ≥ 8)* | *Stringent glycemic control (HbA1c < 7)* | *(aRR) 1.85* | *(0.99, 3.44)* |
| | | *Less stringent glycemic control (HbA1c 7–7.9)* | *(aRR) 1.74* | *(0.88, 3.45)* |
| *Udaykumar P_2022_Clinical Epidemiology and Global Health+* | *Poor glycemic control (HbA1c ≥ 8)* | *Glycemic control* | *0.30* | *(0.04, 2.33)* |
| *Baltas I_2023_The international journal of tuberculosis and lung disease* | *Poor glycemic control (HbA1c ≥ 8)* | *Glycemic control* | *0.66* | *(0.19, 2.25)* |
| *Kim K H_2024_Respirology* | *Controlled glycemic status during TB treatment* | *Uncontrolled glycemic status during TB treatment (RBG ≥ 180mg/dL)* | *(aOR) 1.29* | *0.67–2.35* |
| *Mistry J_2024_Asian J Pharm Clin Res* | *–* | *–* | *Only 65 people with TB-DM. Insufficient to make any meaningful inference* | |
| **Culture non-conversion at 2 months** | | | | |
| Park SW_2012_Eur J Clin Microbiol Infec Dis | Poor glycemic control HbA1c>7) | Glycemic control | 0.62 | (0.14, 2.71) |
| Yoon YS_2017_Thorax | Poor glycemic control HbA1c≥7) | Glycemic control | 0.23 | (0.05, 0.94) |
| **Sputum smear non-conversion at 2 months** | | | | |
| Mi F_2013_TMIHˆ | Poor glycemic control (FPG > 10mmol/L) | Less stringent glycemic control (FPG 7–10 mmol/l) | 1.50 | (0.57, 3.90) |
| | | Stringent glycemic control (FPG < 7mmol/l) | 0.65 | (0.19, 2.15) |
| Nandakumar KV_2013_Plos One | Poor glycemic control (FBS > 100 mg/dl and PPBS/RBS > 140 mg/dl) | Glycemic control | 0.99 | (0.65, 1.51) |
| Mahishale V_2017_Iran JMS | Poor glycemic control (HbA1c ≥ 7) | Glycemic control | 0.12 | (0.06, 0.23) |

@Studies in italics are from updated review from 26 April 2017 to 31 August 2024; *cRR/ aRR/ cOR. aOR data were extracted from the narrative text/ tables or provided by authors, Kornfeld et al provided aRR after adjusting for age, sex, body mass index, use of insulin during treatment, household income, smoking, alcohol consumption; ˆEnd intensive phase and end treatment outcomes reported among those with glycemic status at 2 months and 6 months respectively.



**Table 2. Effect of glucose lowering treatment on unfavorable TB treatment outcomes, summarized as crude and adjusted relative risk (cRR/aRR)@\*.**

| Study ID | Reference group | Exposed group | cRR* | (95% CI) |
|---|---|---|---|---|
| **All unfavorable end (TB) treatment outcome** | | | | |
| Magee MJ_2013_International J of Infectious diseases | OHA only | Insulin | 2.63 | (1.07, 6.46) |
| | | Insulin+OHA | 0.81 | (0.23, 2.80) |
| Vishwanathan V _2014_Journal of Diabetes and its complications ^ | – | – | Insufficient sample to make any meaningful inference | |
| *Kornfeld H_2020_Clinical infectious diseases* | *OHA only* | *Insulin +/- OHA* | *(aRR) 0.47* | *(0.23, 0.98)* |
| *Udaykumar P_Clinical Epidemiology and Global Health_2022^^* | *–* | *–* | *Insufficient sample size to make any meaningful inference* | |
| *Baltas I_2023_The international journal of tuberculosis and lung disease* | *OHA only* | *Insulin +/- OHA* | *0.60* | *(0.08, 4.39)* |
| *Mistry J_2024_Asian J Pharm Clin Res* | *–* | *–* | *Only 65 people with TB-DM. Insufficient to make any meaningful inference* | |

@Studies in italics are from updated review from 26 April 2017 to 31 August 2024;*cRR data extracted from the narrative text/tables or data provided by authors, Kornfeld et al provided aRR relative risk after adjusting for age, sex, body mass index, HbA1C at baseline, household income, smoking, alcohol consumption; ^Unfavorable TB treatment outcomes among those on OHA only, insulin only and both were 0/53, 2/18 and 0/3 respectively. Sufficient outcomes were not there to calculate cRR; ^^Unfavorable TB treatment outcomes among those on insulin +/- OHA and OHA only during TB treatment were 8/70, 0/32. respectively. Sufficient outcomes were not there to calculate cRR.

by Mahishale V et al (2017) found that stringent glycemic control (HbA1c<7%) at baseline, compared to less stringent or poor glycemic control (HbA1c≥7%), resulted in substantial benefits: an 88% reduction in sputum smear non-conversion at 2 months (cRR: 0.12, 95% CI: 0.06, 0.23), a 30% decrease in unfavorable treatment outcomes (aOR: 0.72, 95% CI: 0.64, 0.81), a 2.8-fold higher likelihood of 'non-recurrence' (aOR: 2.83, 95% CI: 2.60, 2.92), and no increased risk of MDR TB [39]. In a Peruvian study by Magee MJ et al (2013), culture conversion was faster among individuals with glycemic control (no criteria specified) compared to those without control, (aHR: 2.2, 95% CI: 1.1, 4.0) [35]. In a South Korean study by Yoon YS et al (2017), culture non-conversion, but not unfavorable treatment outcomes at the end of treatment, was less likely among those with glycemic control (HbA1c<7%) when compared to poor control (HbA1c≥7%) at baseline (cRR: 0.23, 95% CI: 0.05, 0.94) [34].

Chiang CY (2015) et al from Taiwan reported that stringent glycemic control (HbA1c<7%) at baseline was associated with (cRR: 1.89, 95% CI: 1.12, 3.20) higher risk of unfavorable treatment outcome when compared to poor glycemic control (HbA1c≥9%); while less stringent glycemic control (HbA1c 7–8.9%) was not (cRR: 1.53, 95% CI: 0.96, 2.46) [33].

aRR data obtained (effect not included in their published paper) from Kornfeld et al (2020, Chennai, India) revealed that stringent (HbA1c<7%) glycemic control (cRR: 1.85, 95% CI: 0.99, 3.44) and less stringent (HbA1c 7–7.9%) glycemic control (cRR: 1.74, 95% CI: 0.88, 3.45) was not associated with unfavorable outcomes among people with TB-DM when compared to those with poor (HbA1c≥8%) glycemic control [22]. This association was after adjusting for age, sex, body mass index, insulin use during treatment, household income, smoking and alcohol consumption. On request, those with loss to follow up as outcome were included in this analysis. However, those without evaluable data were still excluded (and the extent is unknown). Other studies did not find a significant association (crude and/or adjusted) between glycemic control and TB treatment outcomes [29–32,36–38,40].

## Type of DM treatment and TB treatment outcomes

Six studies examined the effect of DM treatment types on TB treatment outcomes [22,29,30,32,35,41]. In a study from India by Kornfeld H et al (2020), based on information received from the authors, using insulin anytime during treatment had a

**Table 3. Summary of risk of bias in included studies assessed using Modified New Castle Ottawa quality assessment scale for cohort studies @.**

Modified New Castle Ottawa quality assessment scale for cohort studies

| Study ID | Selection (max 4 stars) | Comparability (max 3 stars) | Outcome (max 2 stars) | Overall comment |
|---|---|---|---|---|
| Chiang CY_2015_Plos One | **** | – | ** | Adjusted analysis done but not of the comparison of our interest, glycemic control was not included in the adjusted analysis |
| MiF_2013_TMIH | *** | – | * | No description of derivation of TB-DM glycemic control/uncontrolled cohort, cross sectional data used for analysis, no adjustment, incomplete follow up likely to introduce bias |
| Magee MJ_2013_International J of Infectious diseases | ** | ** | * | Selected group of people with TB-DM (presumptive MDR), only those 70% with TB outcome data included in analysis, adjustment for two confounders only, incomplete follow up likely to introduce bias |
| Nandakumar KV_2013_Plos One | ** | *** | ** | Glycemic control was considered as `known' if FBG/RBG was available on all three occasions, separated by one month and at least one of them was in continuous phase of ATT. Of 667 TB-DM, glycemic status was unknown for 427 (64%). Cut off (FBG < 100mg/dl; RBG < 140mg/dl), for `optimal glycemic control' was not in line with recommendations. (too strict) |
| Park SW _2012_Eur J clin Microbiol Infec Dis | *** | – | * | Excluded extra pulmonary, pulmonary TB with HIV and age. Adjusted analysis not done; incomplete follow up likely to introduce bias |
| Tabarsi P_2014_Journal of Diabetes and Metabolic disorder | **** | – | ** | Adjusted analysis not done |
| Vishwanathan V _2014_ Journal of Diabetes and its complications | *** | – | ** | New smear positive TB cases, adjusted analysis not done |
| Yoon YS_2017_ Thorax | **** | – | * | Adjusted analysis done but for not of the comparison of our interest, incomplete follow up likely to introduce bias |
| Mahishale V_2017_Iran J MS | **** | *** | ** | No risk of bias |
| *Kornfeld H_2020_Clinical infectious diseases* | ** | *** | * | *The study provides limited clarity on 'when' enrolment happened after notification. Lack of clarity about the denominator from where and which date range of notification the study participants were enrolled and potential delay in enrolment (from notification) could result in selection bias and not getting 'true' baseline HbA1C. On request, during treatment lost to follow up were included in the analysis and then cure rate was calculated. Those without evaluable data (not defined and extent unknown) were excluded from the analysis. Ideally as per WHO, these should have been reported and included in the denominator and reported as 'not cured/unfavorable outcome' in the numerator. Operational definition for 'evaluable data' is not mentioned as well. This is likely to introduce selection bias. Though odds ratio was used, on request relative risk (crude and adjusted) was provided. Adjusted analysis was revised to exclude height (as body mass index was already included).* |
| *Udaykumar P_2020_Clinical Epidemiology and Global Health_2022* | *** | – | * | *Insufficient sample size. Study participant loss to follow-up likely to introduce bias, especially considering they were not reported as unfavorable outcome.. On request, authors provided unadjusted analysis after including loss to follow up. Adjusted analysis not done* |
| *Baltas I_2023_The international journal of tuberculosis and lung disease* | *** | – | ** | *Limited sample size for the analysis of interest. Study participants lost to follow-up likely to introduce bias. Adjusted analysis not done for the analysis of interest.* |

*(Continued)*

**Table 3.** (Continued)

| Modified New Castle Ottawa quality assessment scale for cohort studies | | | | |
| --- | --- | --- | --- | --- |
| *Kim KH_2024_Respirology* | ** | ** | *** | *Glycemic control during TB treatment was assessed based on random blood glucose value of 180 mg/dl and study population was classified as control-yes and control-no. Random blood glucose is not a reliable test for glycemic control. Adjusted analysis done. No adjustment done for insulin use.* |
| *Mistry_2024_Asian J Pharm Clin Res* | – | * | ** | *The study provides limited clarity on 'when' enrolment happened after notification (baseline/follow up). No data on patient characteristics. Limited sample size for the analysis of interest.* |

@Studies in italics are from updated review from 26 April 2017 to 31 August 2024

significantly lower risk (aRR: 0.47, 95% CI: 0.23−0.98) of unfavorable end treatment outcomes when compared to not using insulin. This association was after adjusting for age, sex, body mass index, baseline HbA1C, household income, smoking and alcohol consumption. In a Peruvian study by Magee MJ et al (2013), using 'insulin only' had a significantly higher risk of unfavorable end treatment outcomes compared to those receiving 'OHA only' (cRR: 2.63, 95% CI: 1.07−6.47) [35]. Viswanathan V (2014) et al, Udaykumar P (2022) et al, Mistry J (2024) et al from India and Baltas I (2023) et al from the United Kingdom did not have sufficient sample size and did not find any association (see Table 2) [29,30,32,41].

## Discussion

### Summary of findings

This updated review added five new studies and none of them provided additional quality information on the effects of glycemic control and/or type of DM treatment on treatment outcomes among people with TB-DM [22,29–32]. Overall, there is one observational cohort study (2017) with no to limited (not major) risk of bias that provides evidence that stringent glycemic control (HbA1C<7% at baseline) is associated with lower unfavorable end treatment outcomes including recurrence when compared with non-stringent/poor glycemic control [39]. Other studies, though provided some information, had more than one bias or one major bias that limited meaningful inference of findings. We added one study (2020) where the effect of stringent (HbA1C < 7%) and less stringent glycemic control (HbA1C 7–7.9%) and adding insulin during TB treatment was assessed after sufficiently adjusting for confounders [22]. There was no association of stringent glycemic control with TB treatment outcomes. The addition of insulin was protective for unfavorable TB treatment outcomes. These results were not in the published paper but provided by the authors on request [22]. However, major potential selection bias precludes meaningful inference (see Table 3 and S2 Annex).

### Qualitative assessment of the newly added studies

In addition to the quality assessment done for nine studies in the previous review [20], we have added five more studies in the updated review [22,29–32].

The study by Kornfeld et al. (2020) stratified people with TB (not people with TB-DM, n = 389) based on HbA1c 8% cut off and BMI 18.5 kg/m2 cut off into four categories. The TB treatment outcomes were assessed at six months and compared after adjusting for age, sex, height, household income, smoking and alcohol consumption. Surprisingly, among participants with low BMIs, those with poorly controlled diabetes (HbA1c ≥ 8%) demonstrated better TB treatment outcomes. This association persisted even after adjusting for potential confounding variables [22]. Measuring and monitoring insulin resistance levels, could be a potential strategy and this may be explored in future.

The study had some major limitations (see Table 3 and S2 Annex). The study provided limited clarity as to when enrolment happened after notification. There was a lack of clarity about the denominator from where and which date

range of notification study participants were enrolled. Delay in enrolment cannot be ruled out. This could result in selection bias and not getting 'true' baseline HbA1C. During treatment, lost to follow up were excluded (significant extent) from the analysis and then cure rate was calculated (both crude and adjusted analysis). Those without evaluable data (not defined and extent unknown) were excluded from the analysis [22]. Ideally as per WHO, those lost to follow up and not evaluated should be reported and included in the denominator and reported as 'not cured/unfavorable outcome' in the numerator [24].

Kornfeld et al used cOR and aOR as the summary statistic [22]. It is statistically more robust when compared to relative risk. Analysis wise RR should have been used in line with the study design (cohort study). In statistically significant aOR (presented in the published paper) for cure as the outcome, the lower limit of 95% confidence interval was close to one (null value). If RR was used, there is a possibility that these results may not be statistically significant. Height should not have been included in the adjusted analysis as BMI categorized using 18.5 kg/m$^2$ cut off was a key exposure variable (height is part of BMI derivation) [22].

On request, we obtained the results of the effect (aRR) of glycemic control and adding insulin on TB treatment outcomes among people with TB-DM (n = 256) after sufficiently adjusting for baseline confounders and including those lost to follow up, but study participants without evaluable data (extent and definition unknown) remained excluded.

In Udaykumar et al (2022), 102 people with TB-DM were divided into a supervision arm and a non-supervision arm [29]. HbA1c levels were assessed at baseline, 3 months from baseline, and after TB treatment. The study hypothesized that systematically monitoring DM treatment alongside TB treatment through the TB treatment provider would improve TB treatment outcomes [29]. There were two major limitations. Due to limited sample size (n ≈ 100), we were not able to make meaningful inferences for our objective of interest. People with TB-DM lost to follow up were excluded from the analysis. However, the revised cRR analysis results provided to us by the authors included those lost to follow up.

The objective of Baltas et al (2023) was to investigate the hypothesis that DM and poor glycemic control are linked to unfavorable TB treatment outcomes [30]. In a multivariable model, people with poorly controlled DM were not associated with unfavorable TB treatment outcomes (aOR: 0.93, 95% CI: 0.12, 7.29). However, on request, the authors did provide aRR. This result again was limited by a small sample size (n ≈ 100). We were not able to make meaningful inferences for our objective of interest.

During follow up (during TB treatment, not specific when), in a study by Kim KH et al (2024) [31], people with TB-DM (n = 328) were stratified into those with controlled (n = 212) and uncontrolled (n = 116) glycemic status based on random blood glucose level (RBG ≥ 180 mg/dl). Uncontrolled glycemic status was not associated with unfavourable outcomes or TB deaths (all-cause mortality during TB treatment), after adjusting for potential confounders. Random blood glucose is not a reliable test for glycemic control and hence the two arms of glycemic control may not be truly representative. To answer the review's objectives, the authors had HbA1C data at TB diagnosis and probably information on insulin use, but we did not get the required information to include in the review.

J Mistry et al (2024) studied tuberculosis treatment outcomes among people with TB-DM [32]. On request, the authors provided data for both our primary objectives of interest. However, due to limited sample size (n ≈ 65), we were not able to make meaningful inferences from the data.

## Ongoing trials/studies

We summarized the ongoing and completed studies, n = 8 (five from previous review and three from updated review) in S3 Annex [20]. It includes two studies that completed enrolment, two ongoing study and four with unknown status. One of the studies that completed enrollment, published into a paper, that got excluded during full text screening. The other study that completed enrollment was published into three papers, that got excluded during title/abstract and full text screenings. The authors of both these completed studies still have unpublished data for our objectives of interest. We did not get the required information to include in the review.



## Implications of this review and its update

Following this updated review and lack of new quality evidence, the implication are the same as that were published in our previous systematic review (2017) [20]. Among people with TB-DM, there is a dearth of studies with minimal to no risk of bias for the effect of glycemic control and the effect of insulin (with or without OHA), when compared to OHA only, on TB treatment outcomes. There is a need for randomized controlled trials on the effect of glucose lowering treatment options on TB treatment outcomes. As of now, the countries may continue to follow the existing guidelines of DM management among people with TB-DM [15,16]. This also provides an opportunity for the national TB programmes to systematically record glycemic control status (at least cFBG that is feasible in resource constrained routine health system setting) at baseline, intensive phase and continuation phase of ATT. Routine reporting and setting up monitoring mechanisms for the same is the need of the hour. In our previous review (2017) we had also provided a standard methodology involving nine measures (in discussion section under implications) that will ensure comparability among the studies [20]. We would like to add to this the need for adjusting baseline disease severity that is key potential confounder along with glycemic status (where type of DM treatment is the exposure of interest) and type of DM treatment (where glycemic status is the exposure of interest), as applicable (see Table 4 for the 10 measures).

**Table 4. Ten measures* for robustly designed and analyzed cohort studies on the effect of glycemic control and/or type of DM treatment on TB treatment outcomes among people with TB-DM [20].**

| Domain | Ten measures |
|---|---|
| **Sample size** | 1. Conducting a multicenter study that will ensure enough sample size of people with TB-DM |
| **Operational definition** | 2. Using standard diagnostic criteria to diagnose DM [42] and standard definitions for TB treatment outcomes (without excluding those lost to follow up or not evaluated from the study population) [24]. |
| **Exposure group categories** | 3. Use of standard HbA1c (7% and 8%) and cFBG (130 mg/dl and 180 mg/dl) cut offs for glycemic control, thus, providing three arms of glycemia (stringent, less stringent, poor control) [15,16] |
| **Reference category in the exposure group** | 4. Using TB-DM with poor glycemic control as the reference instead of people without DM. Using people with TB-DM on OHAs as the reference instead of people without DM |
| **Longitudinal data collection and analysis** | 5. Assessment of glycemic status, preferably using HbA1c (standardized), at baseline, end of intensive phase and during continuation phase |
| | 6. If three or more cFBG values are available during treatment, considering conversion of mean cFBG during treatment to an estimated HbA1c during treatment [42,43] |
| | 7. A longitudinal data analysis making use of HbA1c or cFBG value at each time point is recommended. This would also adjust for clustering for repeat measurements and not reduce the sample size, usually happens when people with TB-DM are classified based on baseline and follow up HbA1c/cFBG values. This way we can not only analyze the effect of glycemic control on TB treatment outcomes but also the effect of improving or worsening glycemic control with time. |
| **Measure of association** | 8. RR (and not OR) should be used for cohort studies. When compared to RR, OR mathematically overestimates risk. If time of follow up is consistent, then use of RR (cumulative incidence ratios), and if time of follow up is not consistent and the dates are available, then use of RR (incidence density ratios) or HR as the measure of effect |
| **Adjusted analysis** | 9. Adjusted analysis for confounders (age, sex, TB site, TB microbiological status, new or old TB, baseline body mass index, insulin resistance, baseline anemia, human immunodeficiency virus status, steroid use, tobacco and alcohol use) |
| | 10. Adjusting baseline disease severity that is key potential confounder along with glycemic status (where type of DM treatment is the exposure of interest) and type of DM treatment (where glycemic status is the exposure of interest), as applicable |

TB: tuberculosis, DM: diabetes mellitus, TB-DM: People with TB and DM, HbA1c: glycosylated hemoglobin, cFBG: capillary fasting blood glucose, OHA: oral hypoglycemic agents, RR: relative risk, OR: odds ratio; HR: hazard ratio; *nine measures were provided in our previously published systematic review on the same topic [20], tenth measure (number 10) was added in this review



## Conclusion

This systematic review is an update of our previous review published in 2017 where we looked at evidence related to glycemic control (stringent, not stringent, poor) and/or DM treatment (insulin with or without OHAs, OHAs only) on TB treatment outcomes among people with TB-DM [20]. We added five studies between April 2017 and August 2024 in this updated systematic review [22,29–32]. They were not free of risk of bias and therefore did not add to the evidence base [20]. Overall between 1996 and August 2024, with the exception of one study that provided evidence related to glycemic control's protective effect (stringent versus not stringent glycemic control) [39], there is a dearth of well-designed studies on this topic that are free of bias and with sufficient sample size. The methodology related measures provided in this paper regarding observational cohort studies are, therefore, still relevant. Harmonization of the methods is needed and would facilitate comparisons (see Table 4) [20]. To provide more robust evidence than what is available, it is recommended to improve the design and rigor of cohort studies in future research on this topic. We need well-designed randomized controlled trials, specifically for the effect of adding insulin on TB treatment outcomes among people with TB-DM.

## Supporting information

**S1 Annex. Details on the reasons for excluding 93 studies during full-text screening.**
(DOCX)

**S2 Annex. Characteristics of studies included in the review.**
(DOCX)

**S3 Annex. Characteristics of ongoing and completed studies.**
(DOCX)

**S1 Appendix. Search strategy for electronic database search and search results (old, 1996 to Apr 2017).**
(ZIP)

**S1A Appendix. Search strategy for electronic database search and search results (updated, including 26 April 2017 to 31 August 2024).**
(ZIP)

**S2 Appendix. Data extraction form containing information of studies included in the review (old, 1996 to 25 April 2017).**
(ZIP)

**S2A Appendix. Data extraction form containing information of studies included in the review (updated, including 26 April 2017 to 31 August 2024).**
(ZIP)

**S3 Appendix. Prisma checklist.**
(DOCX)

## Acknowledgments

None

## Author contributions

**Conceptualization:** Hemant Deepak Shewade.

**Data curation:** Hemant Deepak Shewade, Prabhadevi Ravichandran, S Satish, S Kiran Pradeep.



**Formal analysis:** Hemant Deepak Shewade, Prabhadevi Ravichandran.

**Funding acquisition:** Hemant Deepak Shewade.

**Investigation:** Hemant Deepak Shewade, Prabhadevi Ravichandran.

**Methodology:** Hemant Deepak Shewade.

**Project administration:** Hemant Deepak Shewade.

**Resources:** Hemant Deepak Shewade.

**Software:** Hemant Deepak Shewade, Prabhadevi Ravichandran, S Satish, S Kiran Pradeep.

**Supervision:** Hemant Deepak Shewade.

**Validation:** Hemant Deepak Shewade.

**Visualization:** Hemant Deepak Shewade.

**Writing – original draft:** Hemant Deepak Shewade, Prabhadevi Ravichandran.

**Writing – review & editing:** Hemant Deepak Shewade, Prabhadevi Ravichandran, S Satish, S Kiran Pradeep, Kathiresan Jeyashree, Preetam Mahajan, Amar N. Shah, Richard Kirubakaran, Ajay M. V. Kumar.

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
