## [Decision Letter · Decision Letter 0]

Dear Dr. Ravichandran,

Thank you for submitting your manuscript to PLOS ONE. After careful consideration, we feel that it has merit but does not fully meet PLOS ONE’s publication criteria as it currently stands. Therefore, we invite you to submit a revised version of the manuscript that addresses the points raised during the review process.

Please submit your revised manuscript by Jul 17 2025 11:59PM. If you will need more time than this to complete your revisions, please reply to this message or contact the journal office at plosone@plos.org . A rebuttal letter that responds to each point raised by the academic editor and reviewer(s). You should upload this letter as a separate file labeled 'Response to Reviewers'.A marked-up copy of your manuscript that highlights changes made to the original version. You should upload this as a separate file labeled 'Revised Manuscript with Track Changes'.An unmarked version of your revised paper without tracked changes. You should upload this as a separate file labeled 'Manuscript'.

We look forward to receiving your revised manuscript.

Kind regards,

Somayeh Momenyan

Academic Editor

PLOS ONE

Journal Requirements:

The authors have read the journal’s policy and have the following competing interests: Hemant Deepak Shewade is a member of the PLOS Global Public Health Editorial Board.

3. Please remove all personal information, ensure that the data shared are in accordance with participant consent, and re-upload a fully anonymized data set.

Additional guidance on preparing raw data for publication can be found in our Data Policy (https://journals.plos.org/plosone/s/data-availability#loc-human-research-participant-data-and-other-sensitive-data) and in the following article: http://www.bmj.com/content/340/bmj.c181.long .

Reviewers' comments:

Reviewer's Responses to Questions

**Comments to the Author**

1. Is the manuscript technically sound, and do the data support the conclusions?

Reviewer #1: Partly

Reviewer #2: Yes

2. Has the statistical analysis been performed appropriately and rigorously?

Reviewer #1: N/A

Reviewer #2: No

3. Have the authors made all data underlying the findings in their manuscript fully available?

Reviewer #1: Yes

Reviewer #2: Yes

4. Is the manuscript presented in an intelligible fashion and written in standard English?

Reviewer #1: Yes

Reviewer #2: No

Reviewer #1: What is the purpose of searching since 1996 again, as it is an updated review, and they already searched until 2017 in their previous review.

Please move the table 1 to supplementary files.

Providing list of Ongoing trials/studies is not necessary, since they did not meet this review's criteria.

In the results section, authors have reported excess of numerical data from included studies, which is not required, they can report it in table is they want to.

Reviewer #2: Abstract

1. It is uncommon to use double parentheses.

2. The names of all databases are not mentioned.

Introduction

3. It is recommended to report the confidence intervals for the specific risk multipliers (e.g., line 78: "two to three times more likely," "four times more likely").

4. The guidelines cited in lines 84–91—are these from the WHO or other organizations?

5. If there is any recent development or ongoing debate regarding insulin versus oral medications for managing TB-diabetes, please provide that information.

6. Since this review updates a previous study, please briefly explain how the evidence has evolved or why new studies might alter the conclusions.

Methods

7. The rationale for including both interventional and cohort studies is not provided, despite their different biases and methodological differences.

8. Do inclusion criteria, such as severity or duration of TB or diabetes, influence eligibility or impact the analysis?

9. To assess bias, please specify whether these evaluations were conducted independently by two reviewers and how disagreements were resolved.

10. The thresholds for heterogeneity (e.g., I² cutoffs) that determine whether to perform a pooled meta-analysis are not specified.

11. You mentioned that "funnel plots are scheduled if enough studies are available" (line 198). Clarifying what "enough" means (e.g., minimum number of studies) would be preferable.

Results

12. The results section occasionally shifts abruptly between different topics (e.g., from study characteristics to specific outcomes). Grouping similar results and providing clear subsections would improve readability.

13. The statement that results are presented qualitatively "due to substantial risk of bias and heterogeneity" could be strengthened by quantifying heterogeneity or explaining the assessment criteria more thoroughly. Clarify how heterogeneity was evaluated.

14. When presenting relative risks (RRs) and odds ratios (ORs), explicitly state whether these are unadjusted or adjusted estimates, as sometimes both are reported but without clarity.

15. For the adjusted analyses (e.g., Kornfeld et al.), specify which confounders were controlled for and discuss how this affects interpretation.

16. Outcomes vary across studies (e.g., sputum conversion, unfavorable outcomes, treatment success/failure). Clearly define each outcome and indicate whether they are directly comparable.

**Do you want your identity to be public for this peer review?** For information about this choice, including consent withdrawal, please see our Privacy Policy

Reviewer #1: No

Reviewer #2: **Yes: ** Farzane Ahmadi

---

## [Author Response · Author response to Decision Letter 1]

25 Jun 2025

Response: We have ensured that our manuscript meets PLOS ONE's style requirements, including those for file naming.

The authors have read the journal’s policy and have the following competing interests: Hemant Deepak Shewade is a member of the PLOS Global Public Health Editorial Board.

Response: An updated Competing Interest statement has been added in the cover letter

3. Please remove all personal information, ensure that the data shared are in accordance with participant consent, and re-upload a fully anonymized data set.

Response: Not applicable, as this study is a systematic review and does not involve individual participant-level data

Response: We have ensured that the reference list is complete and correct. We have added two new references (18 and 19) in the manuscript.

RESPONSE TO REVIEWER COMMENTS

Reviewer #1:

1. Comment: What is the purpose of searching since 1996 again, as it is an updated review, and they already searched until 2017 in their previous review.

Response: Thank you for your query. The same author group was involved in the previous published systematic review and this update. To clarify, this time, we did not search since 1996 again. The previous review was from 1996 to 2017. This search was from 2017 to 2024. We combined the search findings and have presented the same in this updated review. We have reported that this current search was an extension of previously published review (1996 – 25 April 2017) in the methodology section. To make it clearer, we have included the timeline (26 April 2017 to 31 August 2024) in the abstract (line 39-40 of revised manuscript with track changes) and methods in main text (line 190-93 of revised manuscript with track changes) and conclusion (line 723-27 of revised manuscript with track changes) in main text.

According to the Panel for updating guidance for systematic reviews (PUGs), it is advised to include the details of previous years in manuscript.

The definition of “to update” means “to extend up to the present time” or “to include the latest information”.

Moher D, Tsertsvadze A. Systematic reviews: when is an update an update? Lancet 2006;367:881-3. 10.1016/S0140-6736(06)68358-X

2. Comment: Please move the table 1 to supplementary files.

Response: Thank you for your suggestion. The table 1 has been moved to supplementary file and has been renamed as an annex.

3. Comment: Providing list of Ongoing trials/studies is not necessary, since they did not meet this review's criteria.

Response: Thank you for your comment. According to our search strategy registered in the PROSPERO (PROSPERO 2016: CRD42016039101), we intended to communicate with relevant experts in the fields for any trial/research study that is ongoing and recently completed on this topic. We understand that it is best practice to search for unpublished ongoing trials or studies as it helps to include the available evidence and reduce bias. Also, though some of the publications from one of the ongoing trials does not meet the search criteria, there are more publications expected (not yet published) that are in line with our search criteria (see lines 666-73 of revised manuscript with track changes).

Reference

Searching clinical trials registers: guide for systematic reviewers

BMJ 2022; 377 doi: https://doi.org/10.1136/bmj-2021-068791

4. Comment: In the results section, authors have reported excess of numerical data from included studies, which is not required, they can report it in table if they want to.

Response: Thank you for your suggestion. We have tried our best to reduce the numerical data in results narrative. We have provided the numerical data in narrative text only for the key results, other details can be referred to in the tables (as suggested by the reviewer). We think the numerical data that are included in the narrative in the revised manuscript are required for inferring the precision and statistical significance for these key results, therefore, making the narrative text standalone. We hope this is fine.

Reviewer #2:

Abstract

1. Comment: It is uncommon to use double parentheses.

Response: Thank you for your comment. The double parentheses have been removed from the abstract and the main text.

2. Comment: The names of all databases are not mentioned.

Response: Thank you for your comment. We have included Cochrane Database of Systematic Reviews in abstract. Due to restricted word limit in abstract, we did not mention all the sources in abstract. The details regarding these sources have been included in the methodology section of the main text. We hope this is fine.

Introduction

3. Comment: It is recommended to report the confidence intervals for the specific risk multipliers (e.g., line 78: "two to three times more likely," "four times more likely").

Response: Thank you for the comment. In introduction, we have summarized our findings from multiple studies that have been cited. And we have given a approx. whole number that is close to the risk multipliers reported in these studies. Hence, it may not be possible to report the confidence interval as it is different for each effect size reported in each study. We hope this is fine.

4. Comment: The guidelines cited in lines 84–91—are these from the WHO or other organizations?

Response: The targets for glucose control during TB treatment cited in lines 136-43 (of revised manuscript with track changes) is from practical guidance document issued by International Union Against Tuberculosis and Lung Disease (The Union), France, 2019 in collaboration with World Diabetes Foundation. The glycemic control among non-pregnant adults cited in lines 136-43 (of revised manuscript with track changes) is from standards of care in Diabetes issued by American Diabetes Association, 2024. These have been referenced.

5. Comment: If there is any recent development or ongoing debate regarding insulin versus oral medications for managing TB-diabetes, please provide that information.

Response: Thank you for the comment. We have incorporated this. Yes, debate still exists regarding insulin versus oral medications usage in managing TB-diabetes patients. Both have their own advantages and disadvantages. This has been mentioned in lines 144-52 in revised manuscript with track changes.

6. Comment: Since this review updates a previous study, please briefly explain how the evidence has evolved or why new studies might alter the conclusions.

Response: Thank you for the comment. We have incorporated the suggestion. The necessary information has been included in lines 153-66 of revised manuscript with track changes.

Methods

7. Comment: The rationale for including both interventional and cohort studies is not provided, despite their different biases and methodological differences.

Response: Thank you for your comment. Given limited availability of interventional studies (randomized controlled trial, RCT) in this topic, we included both RCT and cohort studies, to generate ample amount of evidence in this under explored area. Both in previous and current review, all studies were cohort and relevant scale (Modified New castle Ottawa scale) has been used to evaluate the risk of bias.

8. Comment: Do inclusion criteria, such as severity or duration of TB or diabetes, influence eligibility or impact the analysis?

Response: We did consider severity of TB and diabetes during inclusion of studies, to an extent, based on available data. We included people on daily or intermittent anti-TB regimens; with or without HIV; managed either in programmatic or clinical (public or private facility) settings; managed in inpatient or outpatient settings; with pulmonary or extra pulmonary TB; with or without cavities at baseline chest radiograph; with microbiologically or clinically diagnosed TB; and with new or retreatment but excluded drug resistant TB. In case of diabetic management, we included studies reporting stringent, less stringent and poor glycaemic control.

However, we did not include studies based on duration of TB or diabetes due to heterogeneity and paucity of relevant information in the screened studies.

We acknowledge that both severity and duration of both TB and diabetes will have effect on the study outcomes and their incomplete reporting represents a limitation.

9. Comment: To assess bias, please specify whether these evaluations were conducted independently by two reviewers and how disagreements were resolved.

Response: Yes, evaluation was conducted independently by two reviewers by using a standardised data consensus excel form and modified new castle Ottawa quality assessment scale. Any disagreements were resolved through discussion and by consulting the third reviewer to reach consensus.

As this aspect is getting repeated in various paragraphs (is applicable for screening, data extraction as well as risk of bias assessment for included studies), we added the following line immediately after “Data Collection and Analysis” section.

“Screening, data extraction, and assessment of bias all were independently performed by two investigators (PR and SKP/ HDS). Any disagreements were resolved through consensus between the investigators, with the involvement of a third investigator (SS), if necessary.”

The same has been mentioned in the lines 230-33 in the revised manuscript with track changes

10. Comment: The thresholds for heterogeneity (e.g., I² cutoffs) that determine whether to perform a pooled meta-analysis are not specified.

Response: Thank you for the comment. As shared in the manuscript “We intended to report the effect separately for interventional and cohort studies using adjusted (aOR, aRR, aHR) or unadjusted (OR, RR, HR) measures along with 95% confidence intervals (CI) and conduct meta-analysis in the absence of clinical, methodological and statistical heterogeneity.”

There are three types of heterogeneity: clinical, methodological and statistical. All three have to be ruled out. Mere presence of clinical and methodological heterogeneity is sufficient to decide to not perform a meta-analysis. If clinical and methodological heterogeneity is not there then one looks at thresholds of ‘statistical’ heterogeneity (I square cut off). In other words, it is not advisable to assess statistical heterogeneity when there is clinical and methodological heterogeneity. As already shared in the manuscript

“Due to the clinical and methodological heterogenicity, we did not perform a meta-analysis”.

“Due to substantial risk of bias and methodological and clinical heterogeneity, we presented the results of these 14 studies qualitatively.”

“As there was only one study without major risk for bias for the review’s primary objective and there was clinical and methodological heterogeneity, a funnel plot to detect publication bias was not generated [38].”

We hope this is fine.

11. Comment: You mentioned that "funnel plots are scheduled if enough studies are available" (line 198). Clarifying what "enough" means (e.g., minimum number of studies) would be preferable.

Response: Thank you for your suggestion. The minimum number of studies for the performance of a funnel plot is justified to be ten. We have clarified this and the line now reads as follows (see lines 273-76 of revised manuscript with track changes)

“If enough studies were included (at least 10) in meta-analysis, we intended to use funnel plots to detect publication bias.”

Results

12. Comment: The results section occasionally shifts abruptly between different topics (e.g., from study characteristics to specific outcomes). Grouping similar results and providing clear subsections would improve readability.

Response:

Thank you. We have incorporated this suggestion. We have clarified the subsections and grouped the similar results to improve readability.

Search findings

Description of included studies

Risk of bias assessment

Glycemic control and TB treatment outcomes

Type of DM treatment and TB treatment outcomes

13. Comment: The statement that results are presented qualitatively "due to substantial risk of bias and heterogeneity" could be strengthened by quantifying heterogeneity or explaining the assessment criteria more thoroughly. Clarify how heterogeneity was evaluated.

Response: Thank you for the comment. We have incorporated this suggestion. Please see lines 311-19 of revised manuscript with track changes.

As shared in the manuscript “We intended to report the effect separately for interventional and cohort studies using adjusted (aOR, aRR, aHR) or unadjusted (OR, RR, HR) measures along with 95% confidence intervals (CI) and conduct meta-analysis in the absence of clinical, methodological and statistical heterogeneity.”

There are three types of heterogeneity: clinical, methodological and statistical. All three have to be ruled out. Mere presence of clinical and methodological heterogeneity is sufficient to decide to not perform a meta-analysis. If clinical and methodological heterogeneity is not there then one looks at thresholds of ‘statistical’ heterogeneity (I square cut off). In other words, it is not advisable to assess statistical heterogeneity when there is clinical and methodological heterogeneity. As already shared in the manuscript

“Due to the clinical and methodological heterogenicity, we did not perform a meta-analysis”.

“Due to substantial risk of bias and methodological and clinical heterogeneity, we presented the results of these 14 studies qualitatively.”

“As there was only one study without major risk for bias for the review’s primary objective and there was clinical and methodological heterogeneity, a funnel plot to detect publication bias was not generated [38].”

Clinical heterogeneity includes differences in populations (pulmonary or extrapulmonary TB, presence or absence of HIV). Methodological heterogeneity includes differences in interventions/ exposure (varying cut-offs and tests to assess glycemic control at different points in TB treatment) across studies. For example, some studies used poor glycemic control as reference (HbA1C more than 8%) while some studies used non-stringent and/or poor control as reference (HbA1C more than 7%). Few clinical studies used presence of absence of cavity as an outcome (not a WHO TB treatment outcome).

This is one of our major discussion points and we have provided a table in the discussion as our recommendation as to how observational cohort studies should be conducted on this topic in future. See Table 4 “Ten measures* for robustly designed and analyzed cohort studies on the effect of glycemic control and/or type of DM treatment on TB treatment outcomes among people with TB-DM [19]”

Thus, meta-analysis was not performed, and results were presented qualitatively.

14. Comment: When presenting relative risks (RRs) a

---

## [Decision Letter · Decision Letter 1]

Effect of glycemic control and type of diabetes treatment on TB treatment outcomes among people with TB-Diabetes: A systematic review (updated August 2024)

PONE-D-24-50252R1

Dear Dr. Ravichandran,

We’re pleased to inform you that your manuscript has been judged scientifically suitable for publication and will be formally accepted for publication once it meets all outstanding technical requirements.

Kind regards,

Somayeh Momenyan

Academic Editor

PLOS ONE

Additional Editor Comments (optional):

Reviewers' comments:

Reviewer's Responses to Questions

**Comments to the Author**

Reviewer #2: All comments have been addressed

2. Is the manuscript technically sound, and do the data support the conclusions?

Reviewer #2: Yes

3. Has the statistical analysis been performed appropriately and rigorously?

Reviewer #2: Yes

4. Have the authors made all data underlying the findings in their manuscript fully available?

Reviewer #2: Yes

5. Is the manuscript presented in an intelligible fashion and written in standard English?

Reviewer #2: Yes

Reviewer #2: Thank you to all the authors who revised all comments and suggestions.

I do not have any further comments.

**Do you want your identity to be public for this peer review?** For information about this choice, including consent withdrawal, please see our Privacy Policy

Reviewer #2: No

---

## [Editor Report · Acceptance letter]

PONE-D-24-50252R1

PLOS ONE

Dear Dr. Ravichandran,

I'm pleased to inform you that your manuscript has been deemed suitable for publication in PLOS ONE. Congratulations! Your manuscript is now being handed over to our production team.

Kind regards,

on behalf of

Dr. Somayeh Momenyan

Academic Editor

PLOS ONE